# Fetal Hippocampal Connectivity Shows Dissociable Associations with Maternal Cortisol and Self-Reported Distress during Pregnancy

**DOI:** 10.3390/life12070943

**Published:** 2022-06-23

**Authors:** Cassandra L. Hendrix, Harini Srinivasan, Integra Feliciano, Justin M. Carré, Moriah E. Thomason

**Affiliations:** 1Department of Child and Adolescent Psychiatry, New York University Langone Health, New York, NY 10016, USA; hs3062@barnard.edu (H.S.); integra.feliciano@nyulangone.org (I.F.); moriah.thomason@nyulangone.org (M.E.T.); 2Department of Psychology, Nipissing University, North Bay, ON P1B 8L7, Canada; justinca@nipissingu.ca; 3Department of Population Health, New York University Langone Health, New York, NY 10016, USA; 4Neuroscience Institute, New York University Langone Health, New York, NY 10016, USA

**Keywords:** fetal programming, prenatal stress, glucocorticoids, resting-state functional MRI, developmental neuroimaging, sex differences

## Abstract

Maternal stress can shape long-term child neurodevelopment beginning in utero. One mechanism by which stress is transmitted from mothers to their offspring is via alterations in maternal cortisol, which can cross the placenta and bind to glucocorticoid receptor-rich regions in the fetal brain, such as the hippocampus. Although prior studies have demonstrated associations between maternal prenatal stress and cortisol levels with child brain development, we lack information about the extent to which these associations originate prior to birth and prior to confounding postnatal influences. Pregnant mothers (*n* = 77) completed questionnaires about current perceived stress, depressive symptoms, and anxiety symptoms, provided three to four salivary cortisol samples, and completed a fetal resting-state functional MRI scan during their second or third trimester of pregnancy (mean gestational age = 32.8 weeks). Voxelwise seed-based connectivity analyses revealed that higher prenatal self-reported distress and higher maternal cortisol levels corresponded to dissociable differences in fetal hippocampal functional connectivity. Specifically, self-reported distress was correlated with increased positive functional coupling between the hippocampus and right posterior parietal association cortex, while higher maternal cortisol was associated with stronger positive hippocampal coupling with the dorsal anterior cingulate cortex and left medial prefrontal cortex. Moreover, the association between maternal distress, but not maternal cortisol, on fetal hippocampal connectivity was moderated by fetal sex. These results suggest that prenatal stress and peripheral cortisol levels may shape fetal hippocampal development through unique mechanisms.

## 1. Introduction

Fetal development is shaped by dynamic processes within the intrauterine environment. The mother supplies oxygen and nutrients to the fetus through the placenta, and in return, fetal metabolic processes act as regulators for placental nutritional composition [1]. The influence of these and other in utero conditions on subsequent postnatal child pathology and behavior is referred to as fetal programming [2]. One model of fetal programming, the cumulative stress model, posits that heightened fetal exposure to prenatal stress contributes to atypical fetal brain structural and functional development, resulting in dysregulation of child physiological processes [3]. Indeed, high exposure to prenatal stress has been linked to increased risk of psychosocial problems and cognitive delays that persist into adulthood [4,5,6]. While these findings clearly demonstrate the long-lasting effects of maternal stress, studying biobehavioral development after birth obfuscates ability to parse out the influence of postnatal factors that also shape child development and that covary with stress during pregnancy (e.g., parenting, environmental stimulation). Studying markers of neurobiological development in the fetus circumvents the issue of controlling for postnatal environmental factors, thereby providing novel insights into the timing and mechanisms of prenatal stress programming.

A primary stress response pathway implicated in fetal programming is the hypothalamic–pituitary–adrenal (HPA) axis. Activation of the HPA axis initiates a hormone release cascade, the end product of which is cortisol, a steroid hormone [7]. Cortisol plays an important role in aiding fetal lung maturation and other developmental processes across gestation; however, high and prolonged exposure to cortisol can have neurotoxic effects. The placental enzyme 2 11β-hydroxysteroid dehydrogenase (11β-HSD2) protects the fetus by converting the majority of maternal cortisol to its inert form, cortisone, as it crosses the placental barrier [8]. Yet, maternal stress can contribute to sustained, high maternal cortisol levels, which in turn can result in 11β-HSD2 insensitivity [9]. When 11β-HSD2 becomes insensitive to cortisol, greater levels of maternal cortisol pass through the placenta and can bind to glucocorticoid receptors in the developing fetal brain [10].

Brain regions rich in glucocorticoid receptors are particularly susceptible to the influences of heightened prenatal cortisol levels. One such region of the brain is the hippocampus, a structure important for learning and memory. Although hippocampal progenitor cells proliferate at moderate levels of cortisol exposure, hippocampal growth and function are impaired at high cortisol levels in multiple species [11,12]. Human MRI studies have linked prenatal stress exposure to reductions in child hippocampal volume [13], slowed hippocampal growth during infancy [14], and weakened hippocampal functional connections with the posterior cingulate cortex, dorsal anterior cingulate cortex, and insula in newborns [15]. Together, such studies provide convergent evidence that exposure to maternal stress and cortisol early in development can impact the developing hippocampus. Questions remain, however, about when in development this programming of the hippocampal network begins, and what patterns are present before birth; such information can shed light on mechanisms underlying stress-induced fetal programming and the intergenerational transmission of psychosocial risk.

Here, we leveraged fetal resting-state functional MRI (rsfMRI) to test the novel hypothesis that maternal stress is associated with altered hippocampal functional connectivity in the fetal brain. To capture both prenatal stress and the proposed mechanism by which it programs fetal brain development, we utilized self-report measures of stress and a direct assessment of maternal salivary cortisol levels during pregnancy. Extensive research in mouse models suggests that male offspring are disproportionately impacted by maternal prenatal stress across multiple systems, in part through sex-specific transplacental signals [16]. Research in humans also suggests that stress-induced fetal programming may be sexually dimorphic, but directionality is mixed, with some studies demonstrating stronger effects for male offspring and other studies suggesting the opposite [17]. A secondary aim of the present analyses was therefore to examine potential moderating effects of fetal sex.

## 2. Materials and Methods

### 2.1. Participants and Procedures

Healthy mothers with singleton pregnancies were recruited from Hutzel Women’s Hospital in Detroit, Michigan during routine obstetrical appointments. Exclusion criteria included age less than 18 years, being a non-native English speaker, or presence of anatomical fetal brain abnormalities during ultrasound and/or MRI examination. Data from this project were obtained during a single study visit at Wayne State University when fetuses were between 22–39 weeks gestational age (GA; *m =* 32 weeks). At this visit, mothers completed self-report questionnaires on current stress, depression, and anxiety, completed a fetal rsfMRI scan, and provided up to 4 saliva samples, with collections separated by 30 min. Manually segmented and quality-assured functional MRI data were available for 165 fetuses at the time of this analysis. From this quality-assured data, fetuses were excluded if they had low birthweight, were born very preterm (<1800 g or <33 weeks GA; *n* = 14), or if they were scanned prior to 25 weeks GA (*n* = 9). We also excluded fetuses with high average motion or few low-motion functional volumes (>1.5 mm max excursion, >0.5 mm mean, rotational: >2° or <100 low-motion volumes, *n* = 22). Lastly, mothers with fewer than 3 saliva samples from the study visit were excluded from analysis (*n* = 41). The final sample for this analysis was 77 mother–fetal dyads who were predominantly Black American and living in low-income households (e.g., 63% of mothers had a household income of less than $20,000). Sample demographics are displayed in Table 1. All study procedures were approved by the Wayne State University Institutional Review Board and written consent was obtained from participating pregnant women.

### 2.2. Measures

#### 2.2.1. Prenatal Distress 

Women completed the *Perceived Stress Scale* (PSS; Cohen et al., 1983 [18]), *Center for Epidemiologic Studies Depression Scale* (CES-D; Radloff, 1977 [19]), and *State-Trait Anxiety Inventory* (STAI; Spielberger et al., 1970 [20]) to assess stress levels, depressive symptoms, and anxiety symptoms, respectively. All questionnaires demonstrated high internal consistency in the present sample (PSS *α* = 0.83; CES-D *α* = 0.98; STAI *α* = 0.87). A composite prenatal distress variable was created based on principal component analysis (PCA), which identified that the PSS, CES-D, and STAI total scores formed one component, which was conceptualized as a maternal prenatal distress factor in line with prior studies (Hendrix et al., 2022 [21]; Thomason et al., 2021 [22]; see Table 2).

#### 2.2.2. Prenatal Cortisol

Up to four maternal saliva samples were collected using Salivettes (Sarstedt, Nümbrecht, Germany) over the course of the two-hour fetal MRI visit, with approximately 30 min between each sample collection. A total of 74 MRI visits took place between 2 pm and 8 pm; however, a small number of saliva collections occurred in the morning (*n* = 4). Time of day was therefore included as a covariate in all cortisol analyses. After collection, saliva samples were stored at −20 °C before being thawed at room temperature, centrifuged at 3000 rpm for 10 min, and assayed in duplicate using commercially available enzyme-linked immunoassay kits (DRG International, Springfield, NJ, USA). Optical densities were read at 450 nm using an Epoch plate reader (BioTek, Winooski, VT, USA), and concentrations were recorded from a calibration curve using a four-parameter logistics curve. Average intra- and inter-assay CVs were 8.9% and 8.5%, respectively.

Cortisol values were winsorized to three standard deviations above the mean to remove biologically implausible values. We calculated two metrics of HPA axis functioning: overall cortisol output during the MRI visit (AUCg), and the change in cortisol concentration across the MRI visit (AUCi; Pruessner et al., 2003 [23]). While AUCg must have a positive value, AUCi may be either negative or positive depending on whether cortisol concentration increased or decreased across saliva samples. AUCg values were log-transformed to correct for skew. Cortisol AUCi did not show evidence of significant skew in the sample and was therefore not transformed.

#### 2.2.3. Fetal fMRI

A 3T Siemens Verio 70 cm open-bore system with a 550 g abdominal four-channel Siemens Flex coil was used to collected fetal resting state functional MRI data. For each participant, 360 axial frames (12 min) of EPI-BOLD data were collected with the following scan sequence parameters: TR = 2000 ms; TE = 30 ms; flip-angle: 80 degrees, slice-gap: none; voxel-size: 3.4 mm x 3.4 mm x 4 mm; matrix-size: 96 × 96 × 25 voxels. 

#### 2.2.4. fMRI Preprocessing

Preprocessing was performed using both FSL (https://fsl.fmrib.ox.ac.uk/fsl/fslwiki/, accessed on 4 April 2022) and Statistical Parametric Mapping (SPM12) software (https://www.fil.ion.ucl.ac.uk/spm/software/spm12/, accessed on 16 February 2022). First, low-motion segments were selected using FSL’s image viewer. Next, Brainsuite was used to manually draw three-dimensional masks around reference images, and these masks were applied to all other volumes within the corresponding low-motion segment [24]. Subsequent preprocessing included reorientation, brain extraction, motion correction, normalization to a 32-week fetal brain template [25], concatenation of volumes across low-motion segments, realignment, reapplication of the fetal brain mask, and spatial smoothing with a 2 mm FWHM Gaussian kernel [26]. Masks were manually drawn for our primary region of interest (ROI), the hippocampus, on the 32-week fetal brain template [25]. The right hemisphere trace was then mirrored onto the contralateral hemisphere as shown in Figure 1.

### 2.3. Analyses

#### 2.3.1. Seed Connectivity Analyses

We performed voxelwise seed connectivity analyses (SCA) using the bilateral hippocampus as our seed. We used the CONN Functional Connectivity Toolbox (v20b; Whitfield-Gabrieli and Nieto-Castanon, 2012 [27]) to create voxelwise bilateral hippocampus resting-state functional connectivity maps for each fetus. Next, SPM12 was used to conduct second-level multiple linear regressions with log-transformed cortisol output (AUCg), cortisol reactivity (AUCi), and prenatal distress as predictors. In all second-level analyses, gestational age at scan, maternal age at scan, whether it was the family’s first or second MRI visit, and time of visit were included as covariates. Resulting t-maps were transformed into enhanced Z maps using probabilistic threshold-free cluster enhancement (pTFCE; Spisák et al., 2019 [28]). pTFCE integrates cluster information to provide voxel-level statistical inference in a probabilistic manner based on Bayes’ rule, increasing sensitivity while also providing appropriate control for false positives. pTFCE is a recommended, developmentally sensitive strategy for improving reliability in the context of multiple comparisons [29]. Connectivity images were subsequently thresholded at *p* < 0.01 (uncorrected), *k* > 20 in line with prior fetal neuroimaging work [30]. Functional connectivity values were extracted from 2 mm radius spheres surrounding the peak of clusters surviving this threshold for subsequent analyses. Extraction was performed using REX [31].

#### 2.3.2. Statistical Analyses

Fetal functional connectivity values extracted from regions of interest that survived cluster correction were imported into SPSS to perform sensitivity analyses. We first examined whether the association between maternal stress and cortisol with fetal hippocampal connectivity remained significant after controlling for additional covariates. Fetal motion during the scan, fetal sex, family income, maternal education, gestational age at birth, and birth weight were selected as covariates based on prior literature [30]. We also examined associations between fetal rotational and translational motion during the scan with our predictors of interest to ensure our effects were not driven by motion. Cook’s D was used to identify potential outliers in all regressions (i.e., values > 1), and analyses were re-run after excluding any outliers. Finally, sex was explored as a moderator of significant associations with fetal hippocampal connectivity by adding a fetal sex interaction term to regression models.

## 3. Results

### 3.1. Associations between Self-Reported Distress and Salivary Cortisol Levels

The distribution of cortisol metrics is displayed in Figure 2. In general, maternal cortisol levels decreased over the course of the visit, consistent with late afternoon and evening diurnal cortisol patterns. Maternal prenatal cortisol output (AUCg) and cortisol reactivity (AUCi) were correlated with each other (*r* = −0.25, *p* = 0.03). However, neither cortisol metric was correlated with self-reported maternal prenatal distress (log cortisol AUCg: *r* = −0.15, *p* = 0.20; cortisol AUCi: *r* = −0.03, *p* = 0.82).

### 3.2. Associations with Fetal Hippocampal Connectivity

We found that greater self-reported distress during pregnancy was associated with greater positive fetal hippocampal connectivity to the right posterior parietal association cortex after controlling for covariates (*β* = 0.44, Δ*R*^2^
*=* 0.17, *p* < 0.001, *95% b*[0.07, 0.21]; Figure 3). No other regions showed altered hippocampal connectivity as a function of self-reported maternal distress at our specified threshold (*p* < 0.01, *k* > 20). Greater prenatal cortisol output (i.e., log-transformed AUCg) was associated with stronger hippocampal connectivity to the dorsal anterior cingulate cortex (dACC; *β* = 0.33, Δ*R*^2^
*=* 0.10, *p* = 0.007, *95% b*[0.05, 0.33]) and to the left medial prefrontal cortex (mPFC; *β* = 0.25, Δ*R*^2^ = 0.06, *p* = 0.04, *95% b*[0.01, 0.35]). Maternal cortisol reactivity (i.e., AUCi) was not associated with fetal hippocampal connectivity at our specified cluster threshold (*p* < 0.01, *k* > 20). The location of significant clusters is displayed in Table 3.

### 3.3. Moderation by Fetal Sex

Fetal sex did not moderate associations between maternal cortisol output and fetal hippocampal connectivity (*hippocampus-dACC RSFC*: *sex X AUCg b* = 0.20, *se* = 0.15, *p* = 0.24, *95% b*[−1.07, 0.27]; *hippocampus-mPFC RSFC*: *sex X AUCg b* = −0.01, *se* = 0.19, *p* = 0.98, *95% b*[−0.39, 0.38]). However, fetal sex did significantly interact with self-reported maternal distress to predict fetal hippocampal functional coupling with the posterior parietal cortex after covariate control (*sex X distress b* = 0.19, *se* = 0.07, *p* = 0.008, *95% b*[0.05, 0.32]). To probe this significant interaction, we examined associations between maternal distress and fetal hippocampal–parietal connectivity separately for male and female fetuses. As displayed in Figure 4, greater maternal self-reported distress was associated with stronger positive hippocampal–parietal functional connectivity for female (*b* = 0.23, *p* < 0.001, *95% b*[0.13, 0.33]), but not male, fetuses (*b* = 0.04, *p* = 0.36, *95% b*[−0.05, 0.14]).

### 3.4. Sensitivity Analyses

Cook’s D values for all analyses were <0.15, indicating that the effects were not unduly influenced by the presence of outliers. Fetal fMRI motion parameters were included as covariates in our primary analyses and were uncorrelated with fetal RSFC metrics as shown in Table 4. We did observe a positive correlation between self-reported maternal distress and number of frames included in the rsfMRI analyses, as well as a positive correlation between cortisol AUCg and translational motion. Relatively high Cook’s D values indicated these correlations were driven by four participants, and removal of these four participants rendered these associations nonsignificant (distress and frame count: *r* = 0.15, *p* = 0.20; cortisol AUCg and translational motion: *r* = 0.22, *p* = 0.06). As an additional sensitivity analysis, we re-ran all analyses without these four participants, and our results were unchanged. It is therefore unlikely that the observed effects are driven by fetal motion.

## 4. Discussion

Extending prior work on associations between maternal stress and offspring hippocampal development, we provide preliminary evidence that the fetal hippocampal network may be sensitive to variation in maternal prenatal stress and cortisol. We found that maternal cortisol levels were related to greater hippocampal functional coupling with frontal regions across both male and female fetuses. Self-reported maternal distress, on the other hand, was related to greater functional coupling between the hippocampus and the posterior parietal association cortex, particularly for female offspring. These dissociable effects persisted after adjusting for covariates and were not driven by outliers.

We observed that greater cortisol output during pregnancy was associated with stronger functional coupling between the hippocampus and two frontal regions rich in glucocorticoid receptors: the dACC and mPFC [32,33,34]. The heterogenous dACC network is implicated in a variety of functions including reward response, emotion regulation, and learning [35,36]. Structural connections between the hippocampus and dACC, comprising the Papez circuit [10,37], mediate contextual fear generalizations [37] and play an active role in memory consolidation [38]. Prior work has found that high glucocorticoid exposure can damage CA3 pyramidal cells, thereby weakening hippocampal anatomical and functional connectivity to the dACC [10,39] and to the mPFC [40]. In contrast, our findings indicate stronger positive hippocampal–frontal connectivity in the context of relatively higher maternal cortisol output. While drastically elevated amounts of cortisol may disrupt hippocampal neuronal growth, moderately high concentrations of maternal cortisol have been linked to *increased* nerve growth factor activity in rat pups [41]. In our sample, higher maternal cortisol levels may still have fallen within a range that contributes to strengthening hippocampal functional connectivity rather than disrupting it. Indeed, overall cortisol elevations are expected throughout the second and third trimester to aid in fetal maturation [42,43]. It is therefore difficult to conclude whether our results are attributable to different directionality at earlier developmental timepoints, or whether they reflect healthy variation in cortisol levels within our sample.

Functional interactions between the hippocampus and mPFC support spatial memory processing [44,45], a cognitive ability *enhanced* by high cortisol secretion among adults [46]. It is possible that moderately high levels of cortisol in the fetal brain accelerate development of the hippocampal network, which in turn can better prepare the offspring to navigate the postnatal environment, and would be highly adaptive in the context of increased threat or high environmental unpredictability. This fits with models demonstrating atypically fast neural maturation in the context of early life stress [47]. An alternative is that altered hippocampal–frontal connectivity is a form of enhanced neural plasticity that renders the offspring brain more malleable to postnatal influences for better and for worse [48]. Future research can explicitly address these possibilities by examining these complex relationships in a prospective, longitudinal framework. Repeated fMRI scans across pregnancy and into the postnatal period would serve to address whether alterations in the fetal hippocampal network are transient or persistent neural phenotypes, and child outcome data would enable better understanding of the implications of these differences for the individual child.

The makeup of our sample is an important consideration in interpretation of these findings. Our sample is predominantly composed of Black American women living in Detroit, a metropolitan area with significant race-based structural inequalities [49], and 63% of these women reside below the national poverty line. In addition to these sources of structural adversity, Black American women experience discrimination at disproportionate rates, which is understood to become biologically embedded and passed across generations [21,50]. These chronic social stressors can lead to blunting of diurnal cortisol patterns among Black American women [51]. It is therefore possible that relatively higher cortisol reflects *lower* exposure to chronic stress or a protective biological adaptation within this group of women.

An additional finding from this study is that maternal overall cortisol output and self-reported acute distress measures did not correlate with one another. Prior work has also found limited or no cross-sectional correlations between perceived stress and salivary cortisol levels in pregnant women from low-to-middle income households [52,53], though these associations can emerge when perceived stress is repeatedly and densely sampled proximal to cortisol collections [54]. A prevailing mechanistic conceptualization of fetal programming is that chronic or severe stress exposure can lead to elevations in maternal cortisol production, which in turn contributes to insensitivity or decreased expression of placental 11ßHSD-2, enabling greater amounts of maternal cortisol to enter the fetal compartment [55,56]. One interpretation of our results is that salivary cortisol levels across a 2 h evening time period is insufficient for comprehensive characterization of stress-related alterations in maternal HPA axis functioning. Alternatively, interindividual variation in perceived stress may program fetal brain development via alternative mechanisms outside of altering maternal peripheral cortisol levels. Future research can examine markers that tap into other points of the proposed HPA axis fetal programming pathway, such as placental 11ßHSD-2 expression or cortisol levels in cord blood. It may additionally be important to explore competing programming mechanisms that potentially interact with maternal–fetal HPA axis functioning, such as alterations in the vagal nerve and autonomic nervous system [57], immune system [58,59], or placental biology [59].

Maternal reported distress was associated with strengthened hippocampal functional coupling with the right posterior parietal association cortex, with a stronger effect for female offspring in our sample. Although we did not have *a priori* directional hypotheses for potential sex effects, several recent review papers describe that glucocorticoid exposure commonly yields more potent programming effects on male offspring [16,59]. Well-replicated preclinical studies in mouse models have isolated sexual dimorphisms in the placenta that lead to this strengthened effect on male neurodevelopment [16,16]. Translating such studies to humans is complex, and the direction of fetal programming-related sex effects varies based on the outcome being studied [17], but it is nonetheless somewhat surprising that we found that female fetuses had more hippocampal network differences than male fetuses. Without additional information about the relevance of this hippocampal difference to future development, it is difficult to interpret this elevated connectivity as a risk or protective response in the brain, or to determine whether it is a transient association without long-term ramifications for child development. Although stress-related neural differences are often conceptualized as harmful in nature, one study found that relatively stronger functional coupling between the hippocampus and posterior parietal cortex in fetuses was prospectively correlated with fewer behavioral and executive functioning problems and greater school readiness at age five [60]. It is therefore possible that our finding of stronger fetal hippocampus–parietal connectivity in the context of acute stress may be an adaptive neural response that prepares offspring to navigate changing environments. Nonetheless, the number of female fetuses in the present sample was moderate (*n* = 30), which points to the preliminary nature of this analysis and the necessity of future replication in independent samples.

## 5. Conclusions

Our results suggest the presence of mechanisms in utero that strengthen hippocampal connectivity in the context of greater prenatal stress and cortisol. Comprehensive approaches to measuring maternal stress and its biological embedding within the same study is thus a necessary direction for future research on fetal programming. Tapping into multiple biological systems uniquely provides the ability to parse distinct, overlapping, and interactive influences on offspring development, which is integral to uncovering the complex mechanisms by which stress can be transmitted from one generation to the next. A strength of our study is examining the intergenerational transmission of psychosocial stress in a group of participants for whom this question is particularly relevant, but who remain severely under-represented in developmental neuroscience research [61,62,63]. The present study represents one small step toward more inclusive, representative science that can advance our understanding of human brain development across a variety of social experiences.

## Figures and Tables

**Figure 1 life-12-00943-f001:**
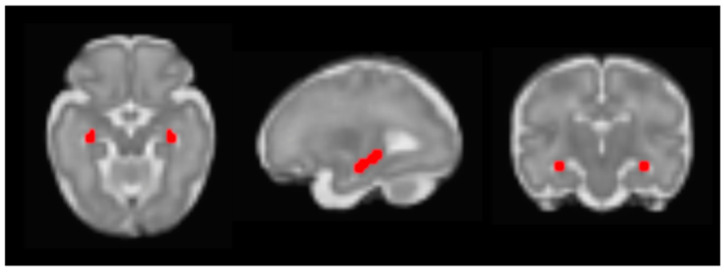
Bilateral hippocampal ROI on a 32-week fetal template.

**Figure 2 life-12-00943-f002:**
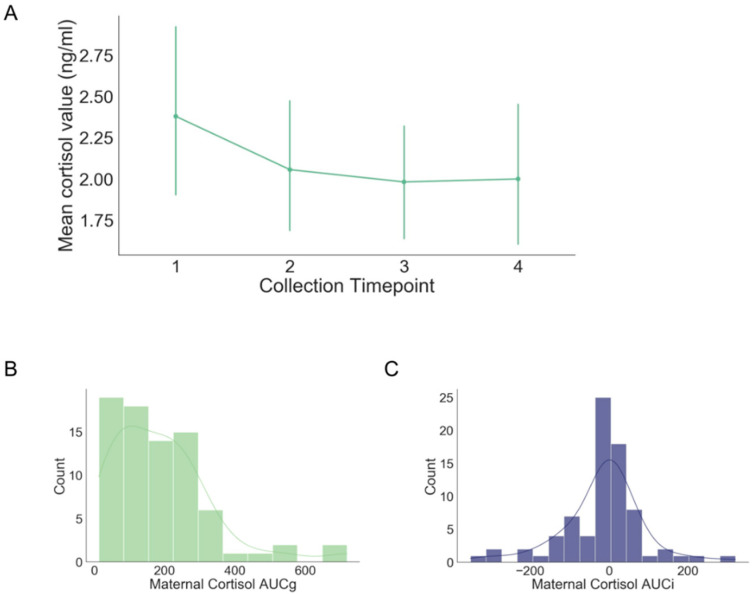
Distribution of maternal salivary cortisol levels. (**A**) Average maternal salivary cortisol levels over the course of the 2 h visit. (**B**) Distribution of maternal cortisol AUCg. (**C**) Distribution of maternal cortisol AUCi.

**Figure 3 life-12-00943-f003:**
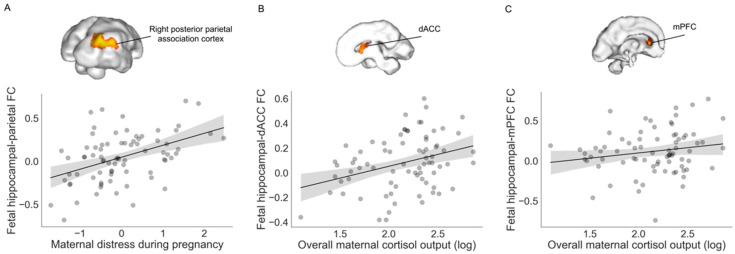
Associations between maternal distress and cortisol during pregnancy with fetal hippocampal connectivity. (**A**) Greater self-reported maternal distress was related to stronger positive functional coupling between the hippocampus and right posterior parietal association cortex at *p* < 0.01 (uncorrected), *k* > 20. (**B**) Higher maternal cortisol output during pregnancy was associated with stronger positive functional coupling between the hippocampus and dorsal anterior cingulate cortex (dACC) and (**C**) left medial prefrontal cortex (mPFC).

**Figure 4 life-12-00943-f004:**
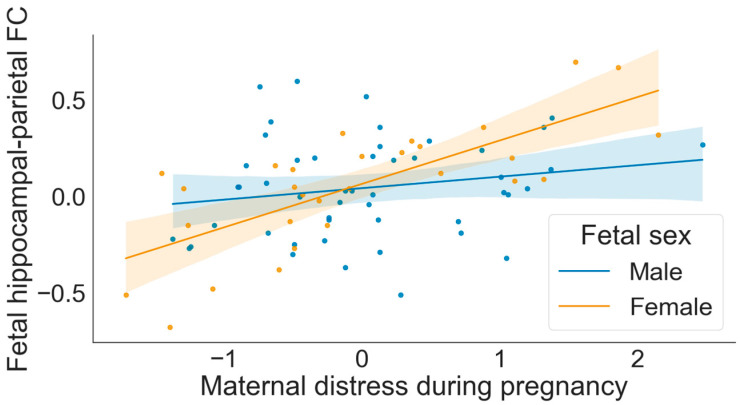
The association between maternal distress and fetal hippocampal connectivity was moderated by fetal sex.

**Table 1 life-12-00943-t001:** Participant Sociodemographics.

	Final Sample (*n* = 77)	Excluded (*n* = 88)	Differences by Group	Final Sample		
				Male (*n* = 48)	Female (*n* = 29)	Differences by Fetal Sex
	*M (SD) or N (%)*		*Stats.*	*M (SD) or N (%)*		*Stats.*
**Sociodemographics**						
Maternal age	25.38 (4.42) years	25.29 (4.83)	*t* = 0.12, *p* = 0.91	26.03 (4.81)	24.32 (3.51)	*t* = 1.80, *p* = 0.08
GA at fetal MRI	32.82 (3.86) weeks	33.17 (3.60)	*t* = −0.49, *p* = 0.62	33.00 (3.87)	32.53 (3.90)	*t* = 0.52, *p* = 0.61
Maternal race			*X^2^* = 4.87, *p* = 0.30			*X^2^* = 4.54, *p* = 0.21
Black	60 (82%)	74 (87%)		36 (78%)	24 (89%)	
White	9 (12%)	5 (6%)		7 (15%)	2 (7%)	
Bi-racial	3 (4%)	4 (5%)		3 (7%)	0 (0%)	
Asian American	1 (1%)	0 (0%)		0 (0%)	1 (4%)	
Other	0 (0%)	2 (2%)		0 (0%)	0 (0%)	
Latina	0 (0%)	0 (0%)		0 (0%)	0 (0%)	
Native American	0 (0%)	0 (0%)		0 (0%)	0 (0%)	
Maternal education	37 (51%) HS diploma/GED or less	50 (59%)	*X^2^* = 1.27, *p* = 0.26	24 (52%)	13 (48%)	*X^2^* = 0.19, *p* = 0.66
Maternal income	43 (63%) < $20,000	52 (69%)	*X^2^* = 0.36, *p* = 0.55	25 (60%)	18 (69%)	*X^2^* = 0.73, *p* = 0.39
Maternal marital status	42 (58%) single	48 (56%)	*X^2^* = 0.02, *p* = 0.90	28 (62%)	14 (52%)	*X^2^* = 1.02, *p* = 0.31
GA at birth	38.97 (1.47) weeks	37.85 (3.28)	***t* = 2.76, *p* = 0.007**	39.25 (1.56)	38.51 (1.19)	***t* = 2.20, *p* = 0.03**
Birth weight	3179.97 (536.95) g	2999.31 (780.99)	*t* = 1.68, *p* = 0.09	3356.25 (540.85)	2888.20 (387.42)	***t* = 4.07, *p* < 0.001**
Fetal sex	29 (38%) female	43 (49%)	*X^2^* = 2.10, *p* = 0.15	--	--	**--**
**rsfMRI characteristics**						
# low-motion volumes	168.14 (51.84)	163.56 (55.46)	*t* = 0.45, *p* = 0.65	167.79 (56.57)	168.72 (43.82)	*t* = −0.08, *p* = 0.94
Mean XYZ translation	0.24 (0.10) mm	0.23 (0.08)	*t* = 0.46, *p* = 0.65	0.22 (0.09)	0.26 (0.11)	*t* = −1.49, *p* = 0.14
Mean PYR rotation	0.40 (0.16) mm	0.39 (0.17)	*t* = 0.46, *p* = 0.64	0.39 (0.15)	0.43 (0.18)	*t* = −0.90, *p* = 0.37
**Maternal prenatal distress and cortisol**						
PSS	15.77 (6.65)	17.13 (6.86)	*t* = −1.24, *p* = 0.22	16.35 (5.83)	14.86 (7.79)	*t* = 0.94, *p* = 0.35
CES-D	15.00 (9.89)	13.77 (9.73)	*t* = 0.80, *p* = 0.43	14.64 (9.65)	15.61 (10.43)	*t* = −0.41, *p* = 0.68
STAI	36.01(8.23)	35.96 (8.87)	*t* = 0.04, *p* = 0.97	36.37 (7.55)	35.45 (9.33)	*t* = 0.47, *p* = 0.64
Cortisol AUCg	191.42 (146.30)	--	--	195.80 (163.15)	184.16 (115.45)	*t* = 0.34, *p* = 0.74
Cortisol AUCi	−15.25 (105.53)	--	--	−14.35 (106.88)	−16.73 (105.10)	*t* = 0.10, *p* = 0.92

*Note.* Sociodemographics for the final sample are displayed above. GA = gestational age, MRI = magnetic resonance imaging. CES-D = Center for Epidemiologic Studies Depression Scale. STAI = State-Trait Anxiety Inventory. AUCg = area under the curve with respect to ground. AUCi = area under the curve with respect to increase. Dyads were excluded from analyses if the fetus was born very preterm, had high motion during the rsfMRI scan, or if mothers provided fewer than 3 usable saliva samples at the fetal MRI visit. Accordingly, excluded dyads gave birth earlier in gestation and had a marginally lower birthweight than included dyads. There were no other differences between dyads who were or were not included in our final analyses. Within the included sample, male fetuses weighed more at birth compared to female fetuses, and there was a marginally significant trend for male fetuses to have a higher gestational age at birth and older maternal age. There were no other sociodemographic differences between male and female fetuses in our final sample. Bolded statistics indicate significant between-group differences at *p* < 0.05.

**Table 2 life-12-00943-t002:** Prenatal distress component loadings.

	Component 1
PSS	0.89
CES-D	0.88
STAI	0.88

*Note.* Together, the PSS, CES-D, and STAI explained 78% of cumulative variance.

**Table 3 life-12-00943-t003:** Differences in the fetal hippocampal network associated with maternal distress or cortisol (AUCg).

	X	Y	Z	Intensity (Fisher Z)	Direction of Effect
**Prenatal Distress**					
Right posterior parietal association cortex	30	−28	6	3.83	Positive
**Prenatal Cortisol AUCg**					
dACC	2	16	−2	2.83	Positive
Left mPFC	−12	24	−6	2.64	Positive

*Note.* We conducted voxelwise group-level seed-based connectivity analyses, seeding in the fetal bilateral hippocampus. We found increased hippocampal to right posterior parietal functional connectivity in the context of increased prenatal distress. We also found increased hippocampal to dorsal anterior cingulate cortex and hippocampal to medial prefrontal cortex functional connectivity in the context of increased overall cortisol output. Results were enhanced using probabilistic threshold-free cluster enhancement and thresholded at *p* > 0.01 (uncorrected), *k* > 20. dACC = dorsal anterior cingulate cortex. mPFC = medial prefrontal cortex.

**Table 4 life-12-00943-t004:** Correlations between fetal fMRI motion parameters and primary variables.

	Number Frames Included in Analysis	Mean XYZ Translation	Mean PYR Rotation
**Maternal predictors**			
Prenatal distress	***r* = 0.23, *p* = 0.04**	*r* = −0.02, *p* = 0.83	*r* = 0.04, *p* = 0.73
Cortisol AUCi	*r = −*0.15, *p* = 0.20	*r = −*0.09, *p* = 0.43	*r = −*0.11, *p* = 0.34
Cortisol AUCg	*r* = 0.14, *p* = 0.21	***r* = 0.25, *p* = 0.03**	*r* = 0.21, *p* = 0.07
**Fetal RSFC metrics**			
Hippocampal–dACC FC	*r* = 0.02, *p* = 0.86	*r* = 0.07, *p* = 0.58	*r* = 0.09, *p* = 0.43
Hippocampal–mPFC FC	*r* = 0.04, *p* = 0.73	*r = −*0.09, *p* = 0.44	*r = −*0.17, *p* = 0.14
Hippocampal–parietal FC	*r* = 0.13, *p* = 0.26	*r = −*0.10, *p* = 0.38	*r = −*0.04, *p* = 0.73

## Data Availability

The data used in analyses for this study are available from the authors upon reasonable request.

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
