# Peer review of "Fetal Hippocampal Connectivity Shows Dissociable Associations with Maternal Cortisol and Self-Reported Distress during Pregnancy"

_life, 2022, doi:10.3390/life12070943_

Round 1
Reviewer 1 Report
Maternal prenatal stress and cortisol levels may affect long-term child neurodevelopment via fetal programming. However, studying biobehavioral development after birth makes it difficult to parse out the influence of postnatal factors. Using a fetal resting-state functional MRI scan, Hendrix et al. conducted Voxelwise seed connectivity analyses to investigate the associations of maternal self-reported distress and cortisol levels during pregnancy with fetal hippocampal functional connectivity. I have a few recommendations and points of clarification below.
1. The authors found fetal sex differences in the association between maternal distress and fetal hippocampal connectivity. Are there any differences in the sociodemographic parameters between male and female offspring? It would be great if sociodemographic data for each gender can also be provided.
2. What is the level of maternal cortisol which is considered high and can disrupt hippocampal functional connectivity? Is there a difference of cortisol level between women with male vs female offspring?
3. It was surprising to see that maternal distress and cortisol levels are associated with positive fetal hippocampal functional connectivity in the current study. Were the other regions of brain checked? Did the authors follow up with these newborns? Did the newborns from women with higher distress levels and/or cortisol levels show the differences in neurobiological development compared with the ones with lower distress/cortisol levels?
Reviewer 2 Report
Congratulations to authors for conducting an elegant study to demonstrate maternal perceived stress and cortisol levels can affect the fetal hippocampus system in a cross-sectional study. A few important points to address to increase the understanding of the study and its implications:
1. Does the cross sectional survey of stress reveal the long-term stress (over 1-2 months) and its effect on the rapidly growing fetal brain? Explain this further.
2. More than 50% of the consented patients were excluded from the study due to various reasons, including those that did not have all four salivary samples. It would help to have supplementary data presented in this large group.
